# Study on the Hot Cracking Law of Inconel 690/52M Welding Material on F304LN Base Metal by Multi-Layer Cladding

**Li Lu [1,2], Zhipeng Cai [1,*], Jia Yang [2], Zhenxin Liang [2], Qian Sun [3] and Jiluan Pan [1]**

[1] Department of Mechanical Engineering, Tsinghua University, Beijing 100084, China; mailluli@163.com (L.L.); pjl-dme@tsinghua.edu.cn (J.P.)
[2] Suzhou Nuclear Power Research Institute Co., Ltd., Suzhou 215004, China; yangjia0554@163.com (J.Y.); liangzhenxin@163.com (Z.L.)
[3] School of Mechanical and Electrical Engineering, Soochow University, Suzhou 215021, China; qiansun@suda.edu.cn
* Correspondence: caizhipeng2021@163.com

**Abstract:** During the welding of 690 nickel-based alloy, solidification cracking (SFC) and ductility-dip cracking (DDC) easily forms, which has a negative effect on the quality of welded joints and service life. The present study examined the effects of welding heat input and cladding layers on the SFC and DDC, as well as their formation mechanism. The microstructure observation, elemental distribution, and Varestraint test were carried out. The results show that SFC and DDC were formed for the Inconel filler metal 52M, and SFC is more prone to form than DDC. The alloy elements such as Fe, Si, C, and P from base metal can expand the solidification temperature range, such that the SFC sensitivity increases. With the increase of welding heat input, the grain size of cladding metal is increased with a great SFC sensitivity. The increasing welding heat input also makes DDC possible due to the formation of a large angle grain boundary.

**Keywords:** 690 nickel-based alloy; solidification cracking; ductility-dip cracking; Varestraint test

## 1. Introduction

Recently, increasing attention has been paid to nuclear energy because it is regarded as an effective alternative to overcome the energy crisis [1–4]. Inconel 690 is a Ni-based alloy, which is mostly employed in nuclear power pressure equipment as a major material due to its characteristics of fatigue performance, creep resistance, mechanical properties, and corrosion resistance at elevated conditions [5–8]. Weld overlay cladding is well known as an excellent method to achieve superior properties in nuclear manufacturing [9–11], because it can generate macroscopic compressive stress on the inner wall of a pipeline to inhibit the expansion of stress corrosion cracking [12–14].

However, hot cracking is a fatal defect for the Ni-based alloy during weld overlay [15,16], which is generated at a high temperature during cooling after welding or during reheating via subsequent welding [17]. Hot cracking consists of solidification cracking (SFC) and ductility dip cracking (DDC). Thus, many studies have been carried out on the hot cracking of Ni-based alloys during weld overlay [18,19]. Zheng et al. [20] argued that intergranular sliding was not responsible for DDC as high temperatures can enhance intergranular sliding. Rapettia et al. [21] studied the effect of composition on DDC of 690 nickel alloy during multi-pass welding. They found that the cracks observed in this study resulted from DDC, not from liquation or SFC. Ahn et al. [17] studied the DDC susceptibility of Inconel 690 using Nb content. Kim et al. [22] studied the effect of a built-up sequence on the hot-cracking resistivity of dissimilar clads. The results showed that hot cracking occurred near the fusion boundary when the 308L was cladded to the buffer layer of Inconel 52. Jelvani et al. [23] focused on the formation behavior of SFC for single laser cladding tracks of a Ni-based alloy. Alizadeh-Sh et al. [24] presented an empirical–statistical approach to predict SFC during laser cladding of a Ni-based alloy on the A-286 Fe-based superalloy.

In this study, Inconel 690 is overlaid on an austenitic stainless steel during welding overlay. The evolution and controlling of cladding metal formation, microstructure, SFC, and DDC is studied, as well as the effect of welding parameters, thermal boundary conditions, and impurity elements on SFC and DDC.

## 2. Materials and Methods

The materials used in the experiment were an austenitic stainless steel F304LN base material and 690 series Inconel Filler Metal 52L M (52M) welding wire produced by Special Metals Welding Products Company, New Hartford, NY, USA. The chemical composition of the base metal and welding consumables are shown in Tables 1 and 2, respectively.

**Table 1.** Chemical composition of F304LN base material (wt%).

| Element | C | Si | Mn | P | S | Ni | Cr | N |
|---|---|---|---|---|---|---|---|---|
| Content | 0.029 | 0.14 | 0.75 | 0.001 | 0.001 | 9.52 | 18.86 | <0.01 |

**Table 2.** Chemical composition of 52M welding wire from SMC (wt%).

| Element | C | Si | Mn | P | S | Ni | Cr | Mo | Al |
|---|---|---|---|---|---|---|---|---|---|
| Content | 0.015 | 0.14 | 0.75 | 0.001 | 0.001 | 59.25 | 30.20 | <0.01 | 0.10 |
| Element | Co | Cu | Ti | Fe | Nb + Ta | | | | |
| Content | 0.001 | 0.04 | 0.22 | 8.04 | 0.84 | | | | |

In690 was overlaid on the F304LN austenitic stainless steel in this study by weld overlay. An F304LN plate with a width of 160 mm and a thickness of 20 mm was used as the base material, and there were grooves with depths ($d$) in the center of the test plate. The cross section of the test plate is displayed in Figure 1, where $d$ = 1, 2, and 3 mm test boards are, respectively, filled with 1, 2, and 3 layers of 52M cladding metal. The filling and welding parameters are listed in Table 3. After weld overlay, weld reinforcement was removed by machining, and a test plate with a size of 160 × 70 × 10 mm was prepared for the Varestraint test, as shown as in Figure 1. There were four samples for each test plate (I#-1 to I#-4, II#-1 to II#-4, and III#-1 to III#-4), as listed in Table 4. The Varestraint test took the center of the test plate (the weld cladding metal) as the research object.

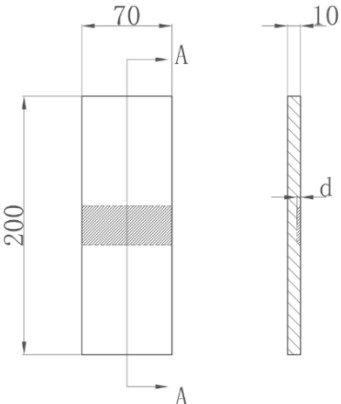

**Figure 1.** Varestraint test board production base material (mm).

The Varestraint test equipment was used to measure the weld hot cracking susceptibility of materials [25,26], which was independently developed by Suzhou Nuclear Power Research Institute Co.,Ltd (Suzhou, China), in this study (Figure 2). During the test, the test plate was fixed in the equipment tooling, and a tungsten inert gas welding (TIG) torch was used to melt the test plates at the specified position. During the remelting process, strain

was applied along the longitudinal weld by tooling to generate hot cracking in the solidification area of the molten pool or heat-affected zone, and the tendency of the weld hot crack was estimated by the crack length. In Figure 2b, $R$ is bending radius, $t$ is the thickness of plate, $\varepsilon$ is strain variable. The cracks on the surface of the test plate were characterized and measured by a Keyence VHX-700F super-depth-of-field three-dimensional microscope.

**Table 3.** Test board welding and filling parameters.

| Current (A) | Walking Speed (mm/min) | Wire Feeding Speed (cm/min) | Base Voltage (V) | Peak Base Value Residence Time (S) |
|---|---|---|---|---|
| 160/120 | 66/110 | 1100/660 | 9.0 | 0.2/0.3 |

Note: Pulse welding is used. In parameter A/B, A is the parameter value corresponding to the pulse peak value, and B is the parameter value corresponding to the pulse base value.

**Table 4.** Varestraint test of samples.

| Test Plate | Dilution Ratio (%) | Sample No. | | | |
|---|---|---|---|---|---|
| I#(1 mm Cladding metal) | ~39 | I#-1 | I#-2 | I#-3 | I#-4 |
| II#(2 mm Cladding metal) | ~15 | II#-1 | II#-2 | II#-3 | II#-4 |
| III#(3 mm Cladding metal) | ~6 | III#-1 | III#-2 | III#-3 | III#-4 |

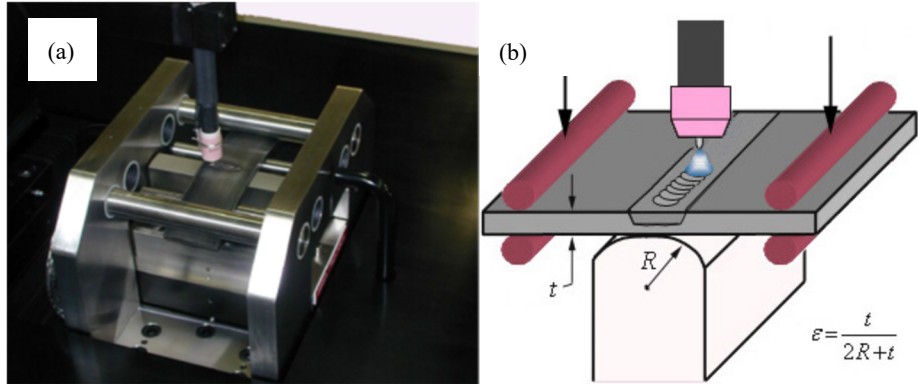

**Figure 2.** Varestraint test: (**a**) equipment Varestraint tooling and (**b**) schematic diagram of the test process.

A SPECTRO-LAB M11 (Kleve, Germany) was used to measure the elemental content in the weld seam. The microstructure and SFC fracture morphology were observed by a field emission scanning electron microscope (FE-SEM) (Sigma300, Zeiss, Jena, Germany) and X-ray energy dispersive spectrometer (EDS). An electron back-scattered diffraction (EBSD) system (Oxford, UK) was used to analyze the crystal orientation and grain boundary angle of the microstructure. The scanning step was 0.3 μm/s. The test results were processed and analyzed using OIM Analysis software.

### 3. Results

*3.1. Varestraint Test Results*

Table 5 shows the macro-morphology of the sample surface after the Varestraint test. It was found that the macro-cracks observed in I# were more than those of II# and III# with the same test parameters. The number and length of hot crackings were formed in I#-1 and II#-2 plates under different currents and the same strain. Compared to the weld of the 150 A welding current, more cracks with greater size were presented in the weld bead of the 200 A welding current. However, it was not visible in the quantity and length of macro-cracks between the III#-1 and III#-2 samples and III#-3 and III#-4 samples for either welding current (150 A and 200 A).

**Table 5.** Crack characterization of the Varestraint test.

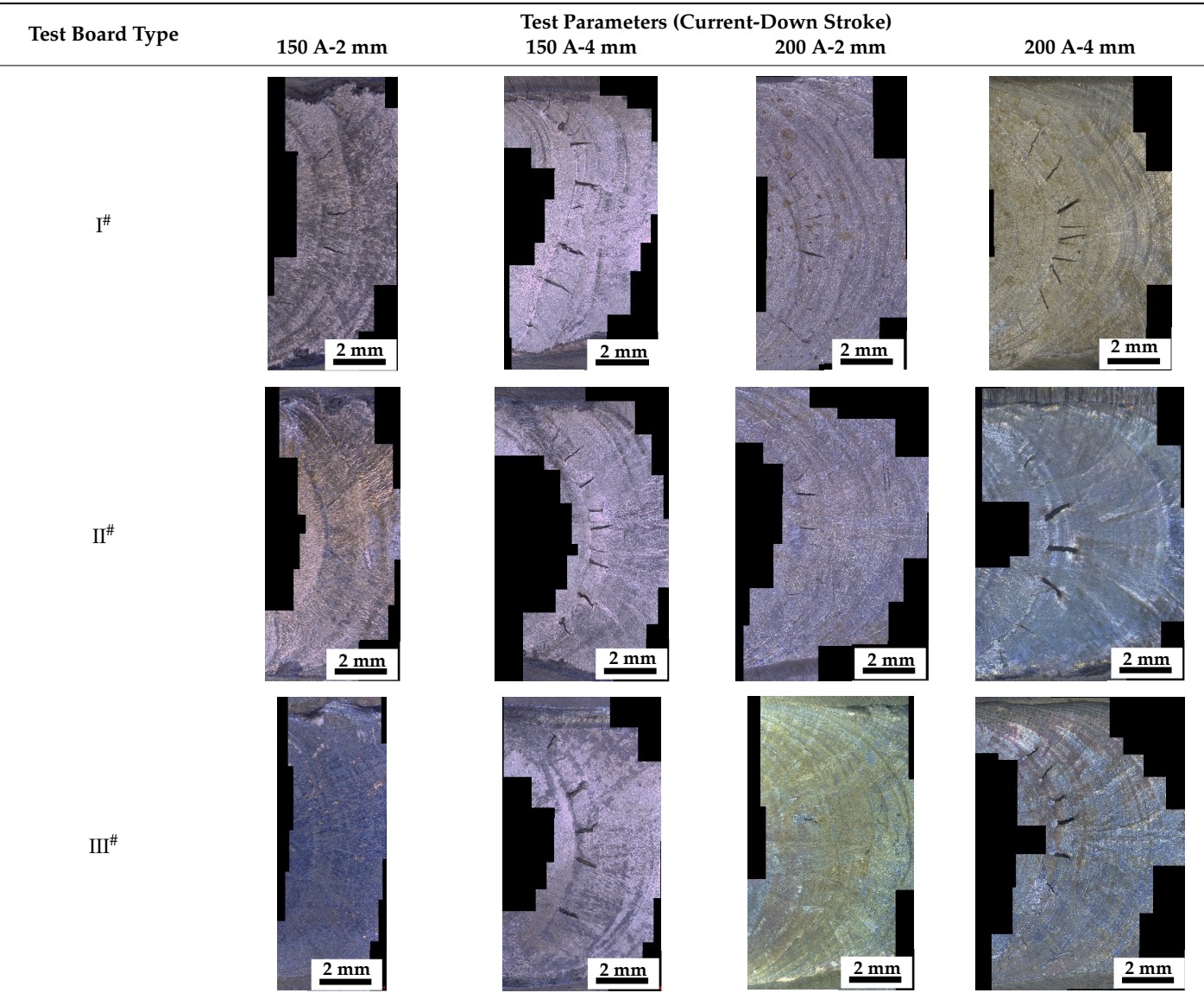

| Test Board Type | Test Parameters (Current-Down Stroke) | | | |
| --- | --- | --- | --- | --- |
| | 150 A-2 mm | 150 A-4 mm | 200 A-2 mm | 200 A-4 mm |
| I# | | | | |
| II# | | | | |
| III# | | | | |

In addition, it can be seen that the cracking characteristics of the II#-4 test plate are very representative. The cracking area of the II#-4 test plate is divided into two parts in Figure 3a. A number of cracks appear between the red curves, which indicate the edge of the molten pool. During the test, this region was undergoing the solidification process with the strain applied, implying that it is a typical SFC initiation place. Conversely, the region between the red curve and blue curve, which is in the heat-affected zone with high temperature, has gone through the solidification process, and a large number of fine straight cracks can be observed. These cracks have the characteristic of a very large aspect ratio, as shown in Figure 4b, which is attributed to the typical brittle DDC.

### 3.2. SFC Statistics, Fracture Morphology

Figure 4 shows the relationship between the crack length of SFC, test current, holding down stroke, and test plate type. The maximum crack size is presented in Figure 5a, and the total crack length (the sum of all SFC crack lengths on a single sample) is presented in Figure 5b. It is found that the maximum crack size is obviously affected by the welding current. When the holding down stroke and the type of test plate were constant, the length of the maximum crack in the weld with the welding current of 200 A was significantly

higher than that of 150 A. In the case of the same welding current and pressing stroke, the crack size was smaller for the sample with more 690 fill layers (I#, I #, III#, test plates were filled with 1, 2, and 3 layers of 690 alloy, and the filling thickness was 1 mm, 2 mm, and 3 mm, respectively). A similar law is observed in Figure 5b: the larger the welding current, the longer the total crack length, while the more 690 filling layers, the smaller the total crack length.

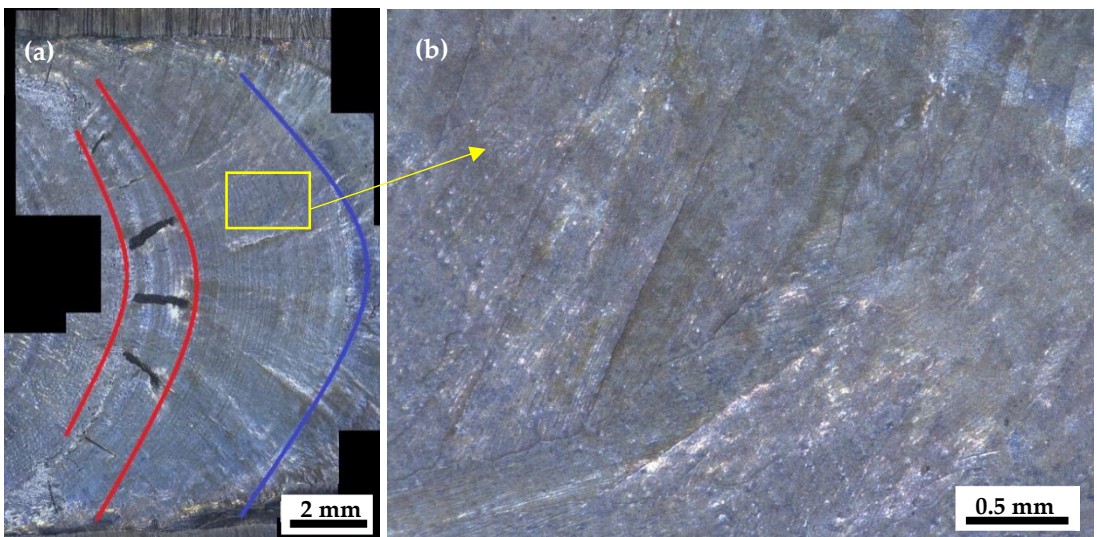

**Figure 3.** Hot crack distribution: (**a**) division of cracking area and (**b**) DDC cracking characteristics.

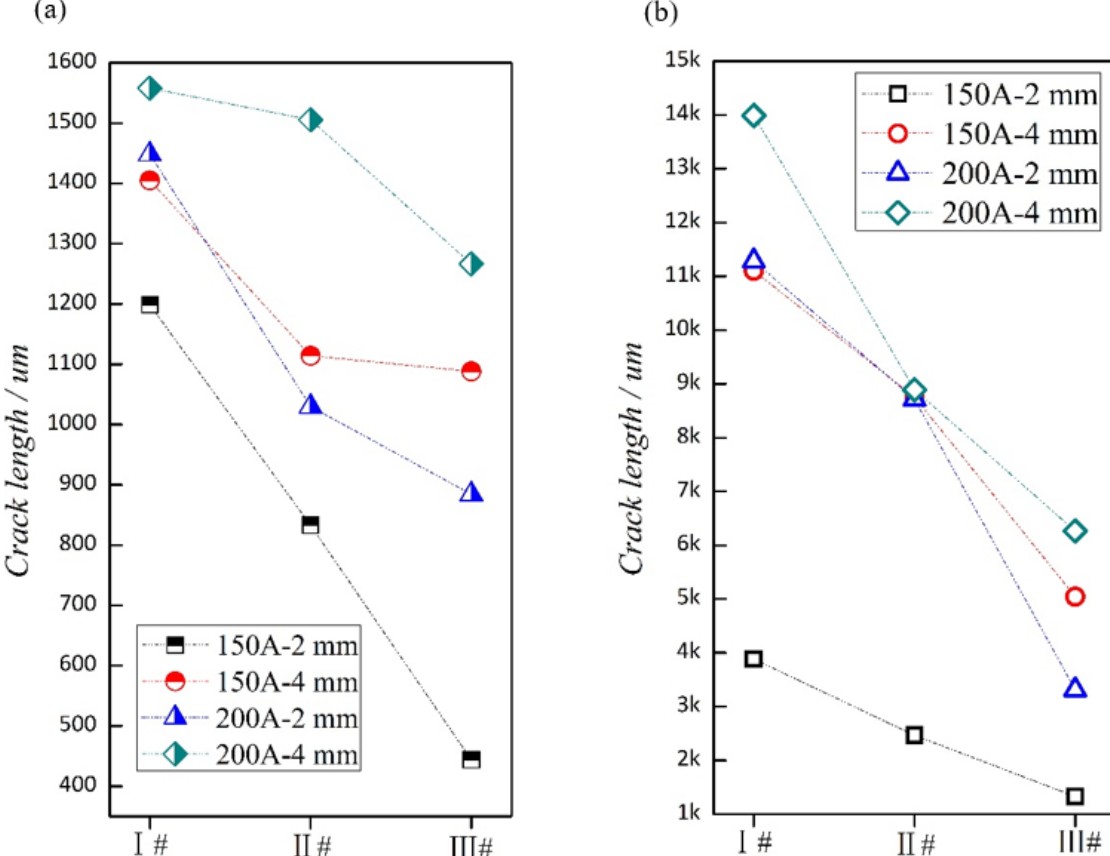

**Figure 4.** SFC crack statistics: (**a**) the maximum crack length and (**b**) the total length of the crack.

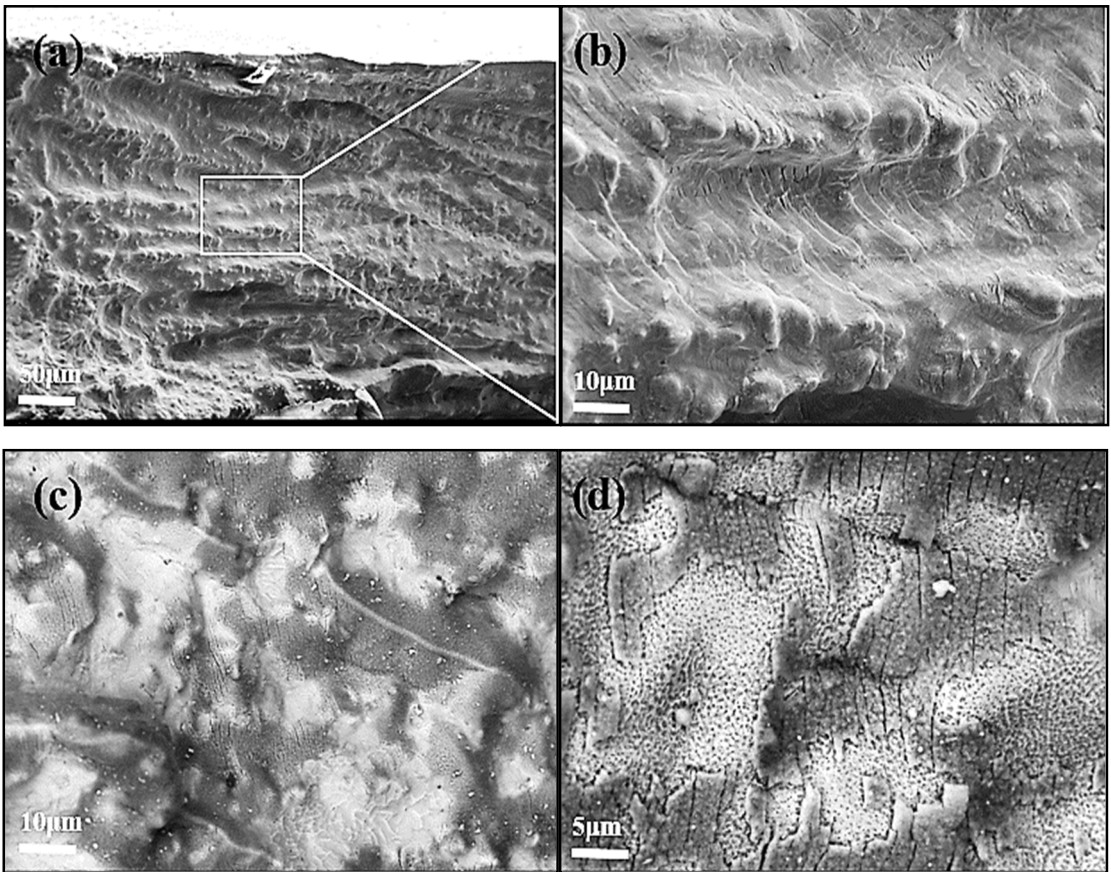

**Figure 5.** Fracture morphology of SFC of nickel-based alloy: (**a**) all-sided fracture surface; (**b**) selected zoom of (**a**); (**c**) liquefaction case 1; and (**d**) liquefaction case 2.

The SFC results on the fracture surface are shown in Figure 5. Intergranular fracture morphology is observed in Figure 5a,b, without any obvious dendritic characteristics and plastic deformation. Furthermore, many liquefaction characteristics are observed in Figure 5c,d. It is inferred that the low-melting-point impurity elements resulted in the formation of SFC. EDS results in Figure 6 and Table 6 show that the elements such as C, O, and K were introduced during the test, and do not exist at the grain boundary crack. Low contents of Ni and Cr were observed, which are the main alloying elements in this study. However, Fe had a high content here. This is due to the dilution of the base metal composition in the weld. In addition, P was an impurity element, with a higher content than the welding material and base metal. The content of Nb was greater than that of the welding material, which is a strong carbide-forming element.

### 3.3. DDC Statistics, Morphology, and Adjacent Grain Orientation

Figure 7 shows the variation trend of DDC crack length with different test currents, holding down strokes, and test plate types. The maximum crack size and total crack length are presented in Figures 7a and 7b, respectively. DDC cracks were not visible for the sample with a 150 welding current and 2 mm holding down stroke (sample No.: I#-1, II#-1, and III#-1), but SFC appeared here. This indicates that SFC is more sensitive than DDC in the welding process. Furthermore, the maximum crack length and total crack length for the case of the 150 A welding current were less than those of the 200 A, which illustrates that the welding current has an important effect on the DDC crack length. Compared to SFC, although DDC is not influenced by the 690 fill layers, the crack length dropped when the filler layers increased to 3 mm.

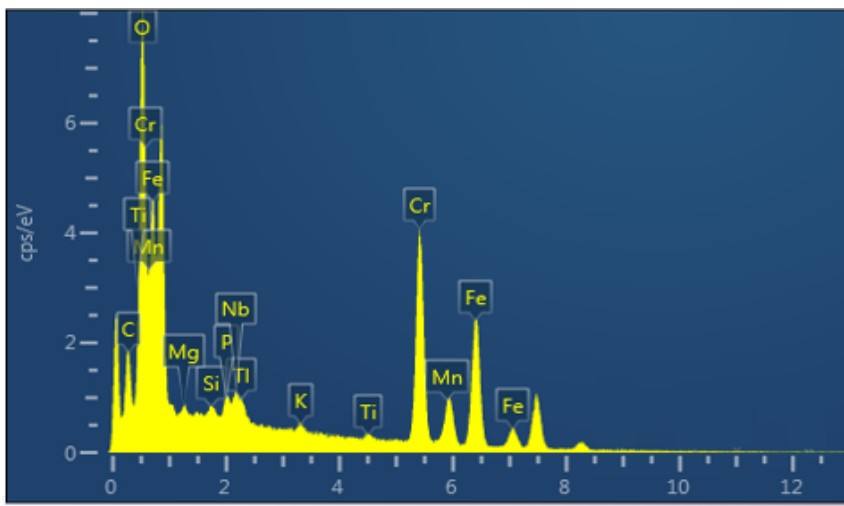

**Figure 6.** Elemental distribution in fracture morphology.

**Table 6.** EDS results in fracture morphology.

| Elements | C | Cr | Fe | Ni | Mn | O | Ti | P | K | Nb | Mg | Si |
|---|---|---|---|---|---|---|---|---|---|---|---|---|
| Fraction (wt.%) | 16.6 | 22.04 | 21.91 | 15.07 | 2.71 | 17.13 | 0.37 | 0.68 | 0.31 | 1.74 | 0.42 | 1.01 |

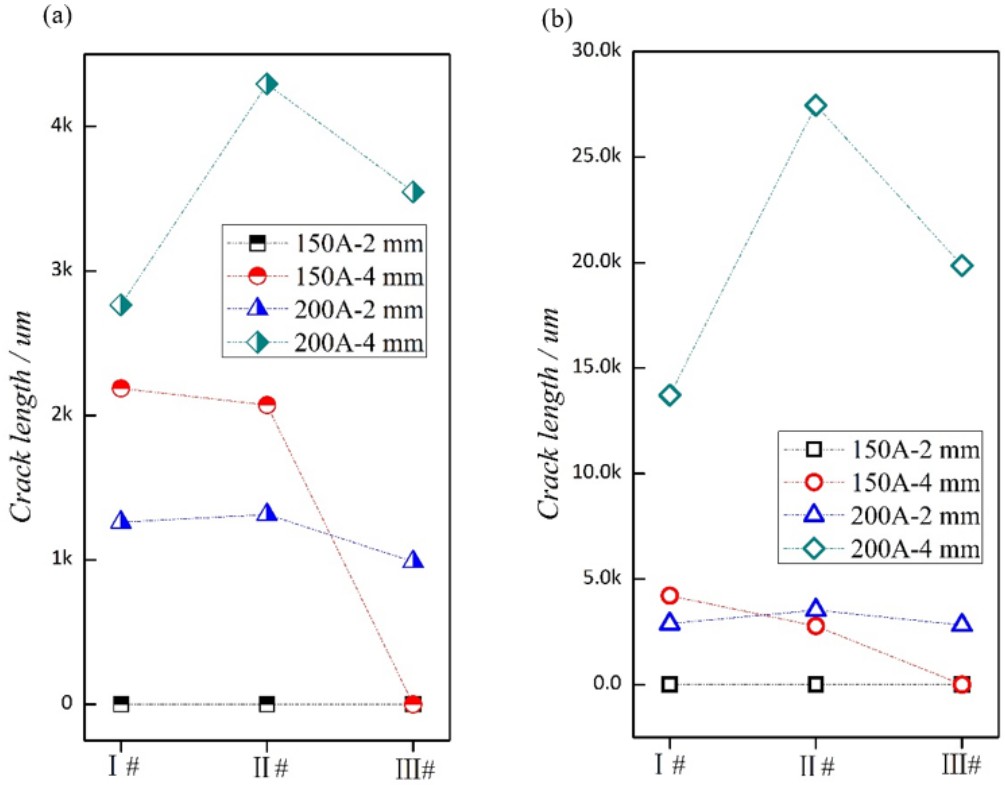

**Figure 7.** DDC crack statistics: (**a**) the maximum crack length and (**b**) the total length of the crack.

In order to further study the relationship between the weld hot cracking and weld microstructure, the typical DDC microstructure morphology and crystal orientation results were observed in this study, as shown in Figure 8. From the secondary electron image of the nickel-based alloy (Figure 8a), the weld hot cracking is clearly observed, consisting of a long crack with a larger width and three small, short cracks; the cracks are essentially initiating and expanding perpendicular to the loading direction. Combining this with

the inverse pole diagram (IPF) of the EBSD analysis results in Figure 8b, the DDC was mainly formed at the columnar grain boundaries (GBs) of the nickel-based alloy. For the nickel-based alloy, the brittle and hard precipitates (such as carbides) were segregated at the grain boundary, resulting in an insufficient plastic reserve. Since deformation of the material was generated at the grain boundary, under the case of large plastic deformation, the brittle and hard grain boundary was prone to DDC. Figure 8c shows the distribution of Euler angles in the EBSD results, and the grain orientation is presented in Figure 9d. The zone marked by white lines in Figure 8c is the position of the grain boundary with DDC, with a significant color difference for the grain. The Euler angle on both sides of the crack indicates a distinct orientation difference and a great grain boundary angle. It is implied that the DDC of nickel-based alloy tends to initiate and propagate at a large angle grain boundary. DDC tends to be produced at the large angle grain boundaries, indicating that large angle grain boundaries with a large atomic mismatch can easily form a segregation of impurity elements and precipitates.

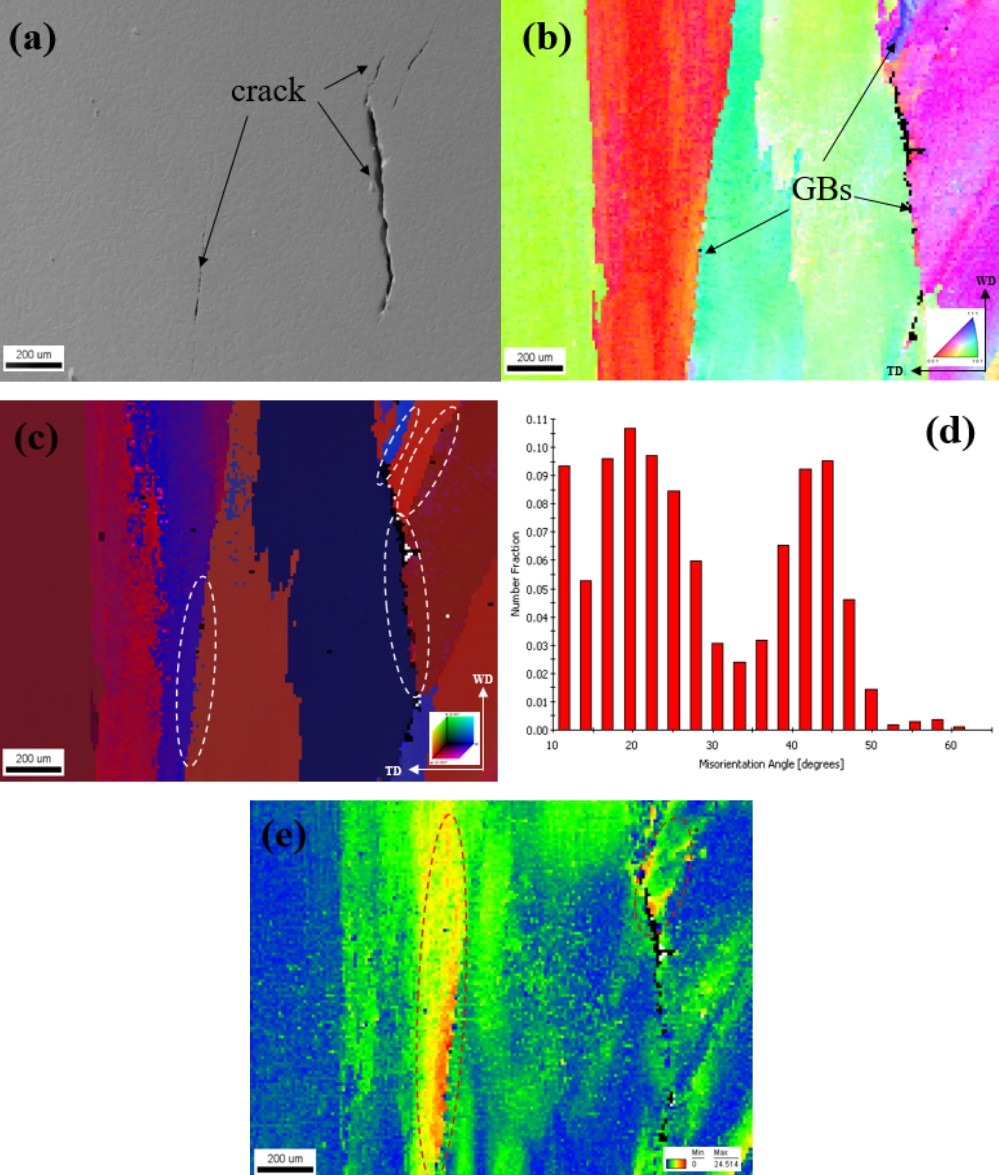

**Figure 8.** Micro-morphology and EBSD results of DDC in nickel-based alloys: (**a**) secondary electron morphology, (**b**) inverse pole diagram IPF, (**c**) Euler angle RGB distribution diagram, (**d**) grain distribution of orientation difference figure, and (**e**) grain distribution of orientation difference distribution map.

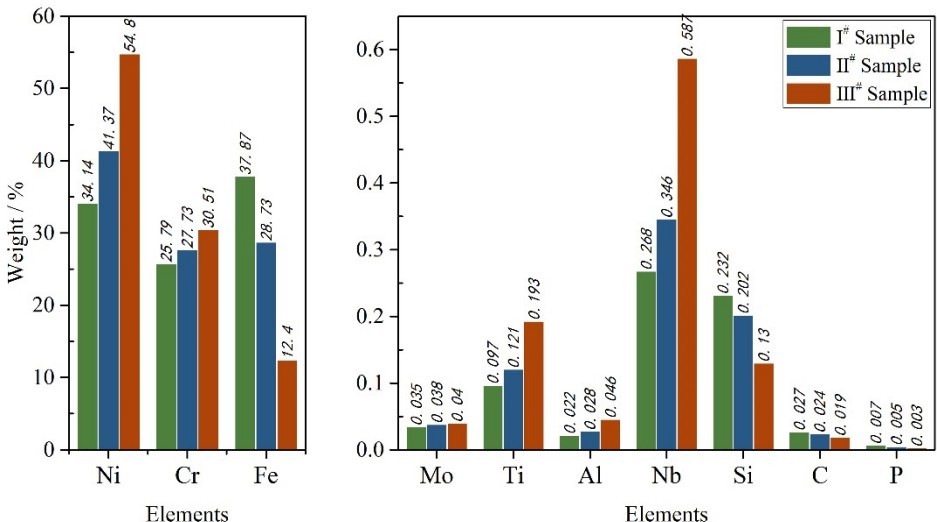

**Figure 9.** The sensitive elements content.

## 4. Discussion

The test results of the alloy composition on the weld surface of the three restraint test plates are listed in Table 7. F304LN base metal melts were melted in the weld during the welding process, resulting in a dilution in the weld composition, which had an effect on the microstructure and intermetallic compound in the weld, as well as the hot crack tendency [27].

**Table 7.** Test results of alloy composition.

| | Alloying Element Content (wt.%) | | | | | | | | |
|---|---|---|---|---|---|---|---|---|---|
| Element | C | Si | Mn | P | S | Ni | Cr | Mo | Al |
| I#Test board | 0.027 | 0.232 | 0.907 | 0.007 | 0.004 | 34.140 | 25.790 | 0.035 | 0.022 |
| II#Test board | 0.024 | 0.202 | 0.894 | 0.006 | 0.004 | 41.370 | 27.730 | 0.038 | 0.028 |
| III#Test board | 0.019 | 0.130 | 0.872 | 0.003 | 0.004 | 54.800 | 30.510 | 0.040 | 0.046 |
| Element | Co | Cu | Ti | Fe | Nb | Ta | | | |
| I#Test board | 0.112 | 0.028 | 0.097 | 37.87 | 0.268 | 0.007 | | | |
| II#Test board | 0.087 | 0.028 | 0.121 | 28.730 | 0.346 | 0.005 | | | |
| III#Test board | 0.040 | 0.026 | 0.193 | 12.400 | 0.587 | 0.003 | | | |

Figure 9 shows the change trend of alloying elements on the weld of the I#, II#, and III# test plates (filled with one, two, and three layers of 690 alloy, respectively). It can be seen that the contents of Ni and Cr in the I# test plate were significantly lower than that of the welding material, but with a higher Fe content. This indicates that the first 690 fill layer had a serious dilution on the F304LN base metal. For the I# and II# test plates, the contents of Ni and Cr in the weld increased as the fill layer increased, but Fe contents decreased, which indicates that the fill layer does not have an obvious effect on the weld composition. Furthermore, the content of Ni, Cr, and Fe in the third layer was very close to that of the welding material, which means that the third layer of the welding bead was hardly affected by the dilution of the base material. Further results show that the contents of Si, C, and P decreased with the increase of the amount of filler layers, indicating that they were introduced into the weld with the dilution of the base metal. The content of Mo, Ti, Al, and Nb increased with the increase of the number of filling layers, since they were the alloy elements for the welding materials. With the increase of the number of filling layers, the base metal dilution had less influence and was similar to the content of the cladding welding material as a result. Thus, the dilution of the base metal was mainly characterized

by the introduction of Fe, Si, C, and P elements into the cladding metal and the reduction of the contents of Mo, Ti, Al, and Nb elements in the cladding metal.

According to the results of the Varestraint test, it can be determined that there are two main factors affecting the hot-cracking susceptibility of 690 nickel-based welding consumables on austenitic stainless steel F304LN. One is the welding current, and the other is the weld composition caused by the dilution of the base material. Figure 10 presents the influence of the welding current on the hot cracking. Comparing Figure 10a and Figure 10d, the bead width and penetration of 200 A were significantly larger than those of 150 A, which means that a larger current may cause a chain reaction that increases the dilution rate. The microstructure magnification of Figure 10a,b is shown in Figures 10d and 10e, respectively. The microstructure consists of austenite for both welding currents, with an inner cellular crystal substructure. However, the grain was smaller and had a more uniform distribution for the case of 150 A, compared to that of 200 A. In Figure 10c,f, the grain size of 150 A is less than that of 200 A. In addition, the 200 A weld had a lot of grains that grew in the horizontal direction. The length of the internal cell was obviously different from that of the normal cell, resulting in the appearance of long, straight grain boundaries. Similar results were not observed in the weld of 150 A. Hence, for the cladding welding of 690 alloy welding consumables on F304LN, a greater heat input causes coarse grains, and the grains and their internal substructures grow along the horizontal direction of the welding, resulting in long and straight grains.

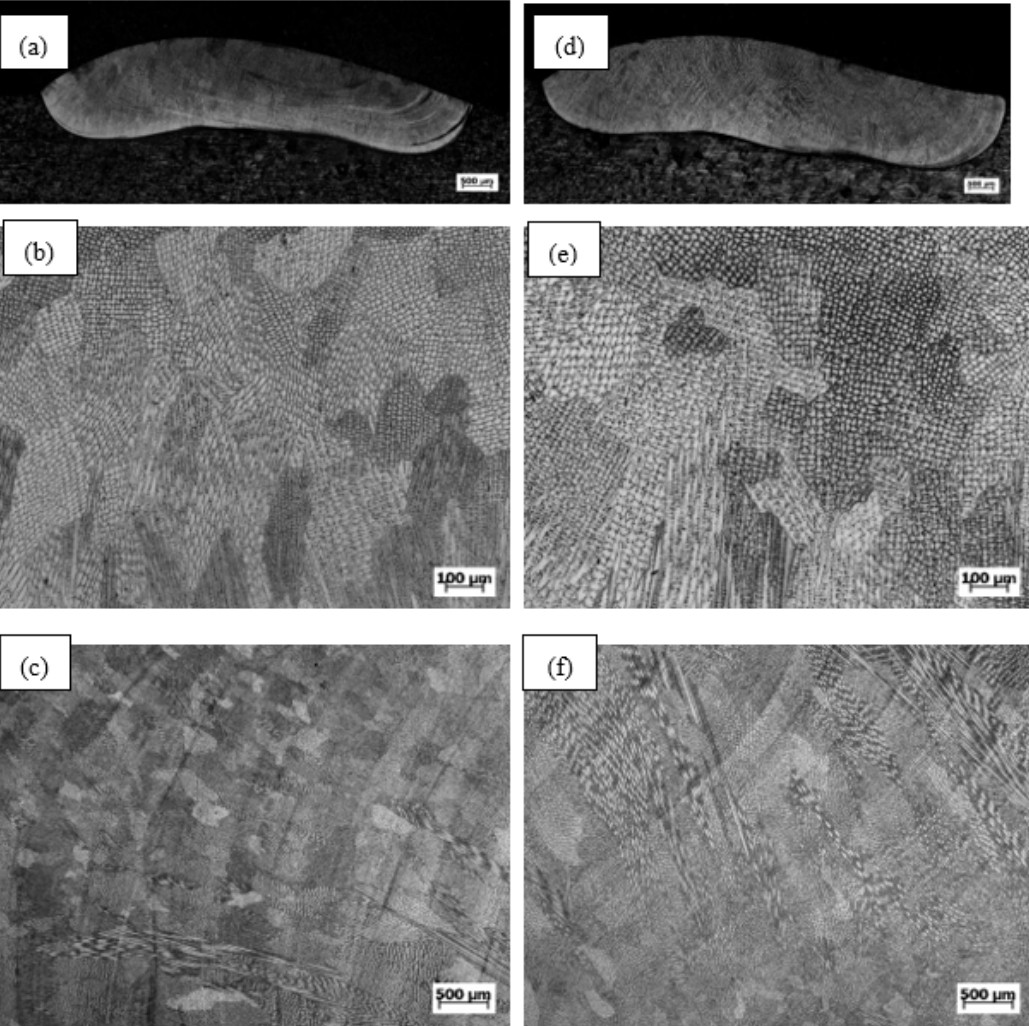

**Figure 10.** Microstructure comparison of 150 A/200 A current cladding metal: (**a**–**c**) 150 A; (**d**) 200 A, macro-section; (**e**) 200 A, micro-section; (**f**) 200 A, macro-surface.

There have been a lot of research results on the influence of elements on the susceptibility of solidification cracking (SFC) of nickel-based alloy welding. Y.J. Shih et al. [28,29] found that Fe, Si, C, and P were key elements in expanding the crystallization temperature range, resulting in a significant increase for the SFC sensitivity and the segregation phenomenon found in the grain boundaries. In this study, the results of the Varestraint test and composition analysis show that the SFC sensitivity of the welds of the I$^{\#}$ plate was the highest, that of the II# plate was the second highest, and that of the III# plate was the lowest; their element contents of Fe, Si, C, and P also sequentially decreased, which is in agreement with the research results of Y.J. Shih et al. [28,29]. It was also verified that Fe, Si, C, and P are key elements that affect the SFC sensitivity. In addition, M.Y. Chen, H.A. Chu, and G. Ko et al. [30–32] found that the eutectic distribution of Ni3 (Nb, Mo) and $\gamma$-NbC on the phase interface was also the key factor leading to SFC. In this study, the Nb observed in the SFC fracture, with a high elemental content compared to that of welding material, also illustrates that Nb is a key element for SFC.

X. Wei, and J.C. Lippold et al. [33–35] studied the crack mechanism of nickel-based welding DDC. They found that Mo, Nb, Ti, etc. were helpful to refine the grain and tortuosity of the grain boundary, which was beneficial to the reduction of DDC sensitivity. In this study, the content of Mo, Nb, and Ti in the weld of I$^{\#}$, II$^{\#}$, and III$^{\#}$ increased in sequence, but the results of the Varestraint test show that the third surfacing layer had a decrease of DDC. It is suggested that increasing the content of Mo, Nb, and Ti to a certain amount has a positive effect on the inhibition of DDC.

## 5. Conclusions

In this study, the solidification cracking and ductility-dip cracking during the welding of 690 nickel-based alloy were studied by elemental distribution, fracture, grain orientation, and microstructure analysis. Solidification cracking (SFC) has more sensitivity than ductility-dip cracking (DDC) for a 690 nickel-based alloy welded on F304L. The positive effect of dilution ratio, heat input, and applied strain on SFC decreased successively. Fe, Si, C, P, and Nb elements in the base metal were the key factors for the formation of SFC. A large heat input caused a greater dilution rate, formation of large-sized grains, and large-angle straight-grain boundaries, resulting in a significant increase in the sensitivity of SFC and DDC. In engineering applications, in order to reduce the hot cracking susceptibility of 690 alloy, it is recommended to select a smaller welding heat input and a larger wire feeding speed for process design under the premise of ensuring the quality of the formed weld bead.

**Author Contributions:** Conceptualization, Z.C. and J.Y.; validation, Z.L. and Q.S.; formal analysis, Z.L.; investigation, L.L.; data curation, L.L. and J.Y.; writing—original draft, L.L.; writing—review and editing, L.L., Q.S., and Z.C.; supervision, J.P. All authors have read and agreed to the published version of the manuscript.

**Funding:** This research was funded by National Natural Science Foundation of China, grant number U20B2031.

**Institutional Review Board Statement:** Not applicable.

**Informed Consent Statement:** Not applicable.

**Data Availability Statement:** Data is contained within the article.

**Conflicts of Interest:** The authors declare no conflict of interest.

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
