# Peer review of "Study on the Hot Cracking Law of Inconel 690/52M Welding Material on F304LN Base Metal by Multi-Layer Cladding"

_metals, doi:10.3390/met11101540_

Round 1

Reviewer 1 Report

This study evaluates the solidification cracking sensitivity that occurs in the cladding layer according to the welding conditions and plate type. Some revisions are suggested before publication.

1. The name of the test that evaluates hot cracking susceptibility is the varestraint test.  Change the variable restraint test to varestraint test

2. Check for mistakes in the names of II # test plates in Table 4. (2 mm Cladding metal)

3. Unify the hot cracking and solidification cracking (SFC) into one in the article.

4. What is the measure of crack length? (SFC or SFC+DDC) Please specify in text.

5. The formation of precipitates such as NbC and Loves in the 690 nickel alloy / F304LN cladding layer affects the generation of SFC. Maybe there is data on the precipitates?

6. Check out the forms for Supplementary Materials, Author Contributions, Funding, Institutional Review Board Statement, Informed Consent Statement, Data Availability Statement, Acknowledgments, and Conflicts of Interest.

Author Response

Review 1#

1. The name of the test that evaluates hot cracking susceptibility is the varestraint test. Change the variable restraint test to varestraint test

Thanks your suggestion. The name of the test that evaluates hot cracking susceptibility had been revised to varestraint test in the revised manuscript.

  1. Check for mistakes in the names of II # test plates in Table 4. (2 mm Cladding metal)

Thanks your suggestion. It has been revised in the manuscript.

  1. Unify the hot cracking and solidification cracking (SFC) into one in the article.

Thanks your suggestion. In this present, hot cracking consists of solidification cracking (SFC) and ductility-dip cracking due to different formation mechanism.

  1. What is the measure of crack length? (SFC or SFC+DDC) Please specify in text.

Thanks your suggestion. The cracks on the surface of the test plate is characterized and measured by Keyence VHX-700F super-depth of field three-dimensional microscope. It has been revised in the manuscript.

  1. The formation of precipitates such as NbC and Loves in the 690 nickel alloy / F304LN cladding layer affects the generation of SFC. Maybe there is data on the precipitates?

Thanks your suggestion. In this paper, the characterization of precipitate phase structure and composition is not done, and the results of literature are cited for correlation analysis. it the future, we will make more work on it. 

  1. Check out the forms for Supplementary Materials, Author Contributions, Funding, Institutional Review Board Statement, Informed Consent Statement, Data Availability Statement, Acknowledgments, and Conflicts of Interest.

Thanks your suggestion. It has been added in the end of the revised manuscript.

Reviewer 2 Report

REVIEW

on article

Study on Hot Cracking Law of Inconel 690/52M Welding Material on F304LN Base Metal by Multi-Layer Cladding

Li Lu, Zhipeng Cai, Jia Yang, Zhenxin Liang, Qian Sun and Jiluan Pan

SUMMARY

Nickel-based alloys are widely used in the nuclear and power industries as corrosion-resistant and heat-resistant materials. Features of the structure allow them to be used at significant stresses and in aggressive environments. Inconel 690 welded joints have shown high sensitivity to hot cracking, especially when welding thick-walled structural elements. The alloys are most susceptible to the formation of ductility dip cracks. In the case of DDS, the critical deformation of crack formation corresponds to about 1.5%.

The authors investigated the welding modes taking into account the susceptibility to hot cracks in the weld. The authors studied the microstructure and morphology of SFC kinks using a field emission scanning electron microscope (FE-SEM) and X-ray energy dispersive spectrometer (EDS).

From this point of view, the article is of interest and is devoted to a topical scientific issue.

The reference list comprises 21 sources.

However, some shortcomings and ambiguities need to be corrected.

COMMENTS.

  1. The English language is not very good, which makes the article less interesting and meaningful. Therefore, I recommend the authors correct English.
  2. The authors must redo the Abstract and bring it in compliance with the requirements of the Metals journal. The scientific problem is not described (Background). The scientific novelty is not indicated. It is unclear what results were obtained. Editors strongly encourage authors to use the following style of structured abstracts, but without headings: (1) Background: Place the question addressed in a broad context and highlight the purpose of the study; (2) Methods: Describe briefly the main methods or treatments applied; (3) Results: Summarize the article's main findings; and (4) Conclusions: Indicate the main conclusions or interpretations. The abstract should be an objective representation of the article.
  3. The references list is poor and comprises only 21 courses.
  4. The Introduction does not reflect current research on the topic of the article. Furthermore, of the 21 references, only 4 are dated after 2018. Therefore, the Introduction must be redone.
  5. The Introduction should summarize the current state of research and why it is important. In the final part of the Introduction, authors must define the purpose of the work and its meaning, including specific testable hypotheses. Make the introduction understandable for scientists working off-topic.
  6. Figure 1 is made careless. Leader lines, extension lines, and dimension lines are of the same thickness.
  7. Please, rephrase paragraph Lines 100-111—a lot of unclear.
  8. Figures 5 and 8 "... pre-bead". I do not understand what it means.
  9. Figures 5 and 8 look messy. I recommend revise them.
  10. Figure 6. Please describe in figures (a), (b), (c), (d).
  11. I recommend the authors revise the Conclusion section following the remarks above.

In general, the article is devoted to an interesting and topical problem studying the welding modes of Inconel 690, considering the susceptibility to hot cracks in the weld. Therefore, I recommend the paper for publishing after corrections.

Author Response

Review 2#

Study on Hot Cracking Law of Inconel 690/52M Welding Material on F304LN Base Metal by Multi-Layer Cladding

Li Lu, Zhipeng Cai, Jia Yang, Zhenxin Liang, Qian Sun and Jiluan Pan

SUMMARY

Nickel-based alloys are widely used in the nuclear and power industries as corrosion-resistant and heat-resistant materials. Features of the structure allow them to be used at significant stresses and in aggressive environments. Inconel 690 welded joints have shown high sensitivity to hot cracking, especially when welding thick-walled structural elements. The alloys are most susceptible to the formation of ductility dip cracks. In the case of DDS, the critical deformation of crack formation corresponds to about 1.5%.

The authors investigated the welding modes taking into account the susceptibility to hot cracks in the weld. The authors studied the microstructure and morphology of SFC kinks using a field emission scanning electron microscope (FE-SEM) and X-ray energy dispersive spectrometer (EDS).

From this point of view, the article is of interest and is devoted to a topical scientific issue.

The reference list comprises 21 sources.

However, some shortcomings and ambiguities need to be corrected.

COMMENTS.

  1. The English language is not very good, which makes the article less interesting and meaningful. Therefore, I recommend the authors correct English.

Thanks your suggestion. The English in this paper has been improved.

  1. The authors must redo the Abstract and bring it in compliance with the requirements of the Metals journal. The scientific problem is not described (Background). The scientific novelty is not indicated. It is unclear what results were obtained. Editors strongly encourage authors to use the following style of structured abstracts, but without headings: (1) Background: Place the question addressed in a broad context and highlight the purpose of the study; (2) Methods: Describe briefly the main methods or treatments applied; (3) Results: Summarize the article's main findings; and (4) Conclusions: Indicate the main conclusions or interpretations. The abstract should be an objective representation of the article.

Thanks your suggestion. The abstract has been revised in the manuscript, as follows:

“During the welding of 690 nickel-based alloy, solidification cracking (SFC) and ductility-dip cracking (DDC) is easy to form, which had a negative effect on the quality of welded joints and service life. This present studied the effects of welding heat input and cladding layers on the SFC and DDC, as well as its formation mechanism. The microstructure observation, elemental distribution, varestraint test are carried out. The results shows that the solidification cracking and ductility-dip cracking are formed for the Inconel filler metal 52M, and the solidification cracking is prone to form compared to ductility-dip cracking. The alloy elements such as Fe, Si, C, and P from base metal can expand the solidification temperature range, such that the SFC sensitivity is increasing. With the increase of welding heat input, the grain size of cladding metal is increased with a great SFC sensitivity. The increasing welding heat input also provides a possibility to the DDC due to the formation of large angle grain boundary.”

  1. The references list is poor and comprises only 21 courses.

Thanks your suggestion. The references list has been improved and 36 courses is involved.

The Introduction does not reflect current research on the topic of the article. Furthermore, of the 21 references, only 4 are dated after 2018. Therefore, the Introduction must be redone.

Thanks your suggestion. The introduction has been redone in the revised manuscript, and the latest references after 2018 has been added.

  1. The Introduction should summarize the current state of research and why it is important. In the final part of the Introduction, authors must define the purpose of the work and its meaning, including specific testable hypotheses. Make the introduction understandable for scientists working off-topic.

Thanks your suggestion. The introduction has been redone in the revised manuscript.

  1. Figure 1 is made careless. Leader lines, extension lines, and dimension lines are of the same thickness.

Thanks your suggestion. A clear Fig. 1 has been instead in the manuscript.

  1. Please, rephrase paragraph Lines 100-111—a lot of unclear.

Thanks your suggestion. It has been revised in the manuscript.

  1. Figures 5 and 8 "... pre-bead". I do not understand what it means.

Thanks your suggestion. It has been revised in the manuscript.

  1. Figures 5 and 8 look messy. I recommend revise them.

Thanks your suggestion. It has been revised in the manuscript.

  1. Figure 6. Please describe in figures (a), (b), (c), (d).

Thanks your suggestion. It has been revised in the manuscript.

  1. I recommend the authors revise the Conclusion section following the remarks above.

Thanks your suggestion. The conclusion has been revised in the manuscript, as follows:

“In this study, the solidification cracking and ductility-dip cracking during the welding of 690 nickel-based alloy is studied, through by elemental distribution, fracture, grain orientation and microstructure analysis. Solidification cracking (SFC) has more sensitivity than ductility-dip cracking (DDC) as 690 nickel-based alloy welded on F304L. The positive effect of dilution ratio, heat input and applied strain on SFC is decrease successively. Fe, Si, C, P and Nb elements in the base metal are the key factors for the formation of SFC. Large heat input causes greater dilution rate, formation of large-size grains and large angle straight grain boundaries, resulting in a significant increase in the sensitivity of SFC and DDC. In engineering applications, in order to reduce the hot cracking susceptibility of 690 alloy, it is recommended to select a smaller welding heat input and a larger wire feeding speed for process design under the premise of ensuring the quality of the formed weld bead.”

In general, the article is devoted to an interesting and topical problem studying the welding modes of Inconel 690, considering the susceptibility to hot cracks in the weld. Therefore, I recommend the paper for publishing after corrections.

Reviewer 3 Report

The submitted manuscript deals with hot cracking of coatings made in the cladding process. I have no major comments. I am in favor of having the manuscript published. Please only pay attention to the following two points.
1. There seems to be a missing verb in the title.
2. Figs. 2a and 2b are unnecessary. Better include a process scheme

Author Response

Review 3#

  1. There seems to be a missing verb in the title.

Thanks your suggestion. The title has been revised as “Study on Hot Cracking Law of Inconel 690/52M Welded on F304LN Base Metal by Multi-Layer Cladding”

  1. Figs. 2a and 2b are unnecessary. Better include a process scheme

Thanks your suggestion. Figs. 2a has been removed. Fig. 2b is Equipment variable restraint tooling, which is corresponding to Fig.2c, so it needs be retained.

Reviewer 4 Report

The manuscript entitled “Study on Hot Cracking Law of Inconel 690/52M Welding Material on F304LN Base Metal by Multi-Layer Cladding” dealing with cladding additive manufacturing has been reviewed.

  1. Please clarify if you found the values of Table 1 and 2 by chemical tests or you found it from the manufacturer data sheet?
  2. Remove the wording “Note: Pulse welding is used. In parameter…” from Table 3. Please place it below this table.
  3. Figure 2a shows nothing. Replace or remove it.
  4. Please provide more discussions on the figures presented in Table 5.
  5. Improve the quality of the wording in Figure 4. The scale bar is not readable.
  6. Change the title of section 4 to “Discussions”.
  7. Update the introduction with the following new references.

  • Ke, Y. and J. Xiong, Microstructure and mechanical properties of double-wire feed GTA additive manufactured 308L stainless steel.
  • Dai, F., H. Zhang, and R. Li, Process planning based on cylindrical or conical surfaces for five-axis wire and arc additive manufacturing
  • Tang, S., G. Wang, H. Song, R. Li, and H. Zhang, A novel method of bead modeling and control for wire and arc additive manufacturing.
  • Fang, X., C. Ren, L. Zhang, C. Wang, K. Huang, and B. Lu, A model of bead size based on the dynamic response of CMT-based wire and arc additive manufacturing process parameters.
  • Kulkarni, J.D., S.B. Goka, P.K. Parchuri, H. Yamamoto, K. Ito, and S. Simhambhatla, Microstructure evolution along build direction for thin-wall components fabricated with wire-direct energy deposition.

Author Response

Review 3#

The manuscript entitled “Study on Hot Cracking Law of Inconel 690/52M Welding Material on F304LN Base Metal by Multi-Layer Cladding” dealing with cladding additive manufacturing has been reviewed.

Please clarify if you found the values of Table 1 and 2 by chemical tests or you found it from the manufacturer data sheet?

Thanks your suggestion. The chemical composition in table 1 and 2 was provided by the manufacturer, and we make it to reconfirm.

Remove the wording “Note: Pulse welding is used. In parameter…” from Table 3. Please place it below this table.

Thanks your suggestion. It has been revised in the manuscript.

Figure 2a shows nothing. Replace or remove it.

Thanks your suggestion. It has been removed.

Please provide more discussions on the figures presented in Table 5.

Thanks your suggestion. It has been removed. The cracks in Table 5 are counted in detail and drawn into figures 5 and 8 for detailed discussion.

Improve the quality of the wording in Figure 4. The scale bar is not readable.

Thanks your suggestion. It has been revised in the manuscript.

Change the title of section 4 to “Discussions”.

Thanks your suggestion. The title of section 4 has been changed to “Discussions” in the revised manuscript.

Update the introduction with the following new references.

Thanks your suggestion. The following references has been updated in the revised manuscript.

Ke, Y. and J. Xiong, Microstructure and mechanical properties of double-wire feed GTA additive manufactured 308L stainless steel.

Dai, F., H. Zhang, and R. Li, Process planning based on cylindrical or conical surfaces for five-axis wire and arc additive manufacturing

Tang, S., G. Wang, H. Song, R. Li, and H. Zhang, A novel method of bead modeling and control for wire and arc additive manufacturing.

Fang, X., C. Ren, L. Zhang, C. Wang, K. Huang, and B. Lu, A model of bead size based on the dynamic response of CMT-based wire and arc additive manufacturing process parameters.

Kulkarni, J.D., S.B. Goka, P.K. Parchuri, H. Yamamoto, K. Ito, and S. Simhambhatla, Microstructure evolution along build direction for thin-wall components fabricated with wire-direct energy deposition.

Round 2

Reviewer 1 Report

Delete the part that is not related to the content at the end of the manuscript.

Reviewer 2 Report

The authors revised the article in accordance with the comments. The article looks much better. I ask the authors to pay attention:

The last reference has no number: J.C. Lippold, N.E. Nissley. Further investigations of ductility-dip cracking in high chromium, Ni-base Filler Metals. Welding in the world, 2007, 51, 24-30. 

Reviewer 4 Report

This paper is ready to publish.